# Influence of Specimen Thickness on the Acquisition of Al6061-T6 Material Properties Using SHPB and Verified by FEM

**DOI:** 10.3390/ma14010205

**Published:** 2021-01-04

**Authors:** Yeon-Bok Kim, Jeong Kim

**Affiliations:** Department of Aerospace Engineering, Pusan National University, Geumjeong-gu, Busan 46241, Korea; 941208b@pusan.ac.kr

**Keywords:** Split-Hopkinson pressure bar, material properties, high strain rate, finite element method, influence of specimen thickness, transmitted pulse, reflected pulse, impedance

## Abstract

The Split-Hopkinson pressure bar (SHPB), which is used for acquiring material properties at high strain rates (10^2^–10^4^ s^−1^), requires proper specimen size selection. Under the same applied pressure, an increased S-S curve is obtained as the thickness of the specimen decreases. In this study, 1.5 t, 2.0 t, 3.0 t, 5.0 t, and 7.0 t specimens of Al6061-T6 material were tested under 1.0 bar to understand the influence of specimen thickness on the acquisition of material properties. To grasp the behavior of the SHPB test in real time, Finite Element Method (FEM) was performed using the LS-DYNA program. During the SHPB test, the impedance is increased due to the variation in the specimen area. Because of the influence of impedance, the transmitted pulse increases, and the reflected pulse decreases. As a result, the specimen is deformed in the high-strain rate region, and the S-S curve is increased as the thickness decreases. In addition, by performing the test under different pressure conditions that created similar strain rate regions, the material properties remained constant with thickness variations.

## 1. Introduction

Researchers and engineers have many things to consider when manufacturing product. Among them, selection of the material for the product is basic and should be carefully considered. Researchers should select a material in consideration of the environment in which the product will be applied and grasp the characteristics of the material. The material properties are affected by strain rate effect and different in the quasi-static and dynamic [1]. When the high-speed impact incidents such as an aircraft or a car crash is considered, both the quasi-static data and dynamic material properties are required. Currently, the machines for obtaining high-speed material properties are typically high-speed tensile test machine [2], Split-Hopkinson pressure bar (SHPB) [3], and Split-Hopkinson tensile bar (SHTB) [4].

The SHPB could acquire dynamic material properties at strain rates from $10^{2}\sim 10^{4}$/s. It was developed from the initial model [5,6] and studied without violating the 1D wave equation and SHPB theory. It acquires the material properties of the specimen by using elastic waves with different impedance between the pressure bar (composed of the incident bar, reflected bar, and transmitted bar) and the specimen. Previous studies conducted acquisition of material properties using specimen of various materials such as concrete [7], composite [8], rubber [9,10] and metal [11,12,13,14].

In SHPB test, characteristics and shape of the specimen should be considered. Impedance is an inherent value and different by each material. Shin [15] conducted a study on the principle of stress wave propagation according to impedance in different medium. Because the pressure bars used in the SHPB should behave elastically during the test, high-strength steel is used. The high-strength steel has generally high impedance value. If the impedance of the specimen compared to the pressure bar is too low, the transmitted pulse is difficult to measure. Chen [16] and Johnson [17] studied the material of the pressure bar from steel to aluminum to lower its own impedance. In other words, it is necessary to select the material of pressure bar according to the type of specimen for each test.

In general, the specimen size in the quasi-static test should comply with the ASTM standard, but there is no a specific standard for the specimen size used in SHPB experiment. It is recommended to use the appropriate specimen diameter and thickness by referring to previous studies [18,19]. Pankow [20] studied different L/D ratio (length L divided by diameter D) along with specimen shape. Wang [21] conducted different outside diameter of specimen. Previous studies [20,21] suggested that different material properties can be obtained by depending on the size of the specimen. Since the specifications of the SHPB possessed by each laboratory differ, the size of the specimen should be changed accordingly. Therefore, it is important to understand the influence of the specimen thickness through SHPB test by thickness.

First, we selected Al6061-T6 as the material used for SHPB. Aluminum is lighter than other metals, has excellent heat conduction, and has advantage of easy processing. Among the Aluminum series, Al6061-T6 is used in automobile and aircraft fields and applied in high-speed situations [22,23]. It means that researching the acquisition of dynamic material properties of A6061-T6 is required.

In the case of SHPB test, it is difficult to understand the specimen information in real time except for the wave pulse because the elastic wave is measured in milliseconds. The finite element method (FEM) is possible to predict the behavior of objects quickly and accurately in specific environments. The analysis of the known methods of solution of the problem are possible only in the simplest cases, but the solution is obtained by using FEM in more complicated cases [24]. It offers advantages in terms of cost and time it takes to build and study a real environment. Therefore, an SHPB analysis model was built using the numerical analysis program, LS-DYNA.

The objectives of this investigation are to understand the influence of the SHPB’s specimen thickness on the acquisition of material properties. In this study, tests for each thickness (1.5 t, 2 t, 3 t, 5 t, and 7 t) under the same pressure conditions (1.0 bar) were performed using Al6061-T6. We verified by FEM to compare the analysis results and test results.

There are many physical quantities in this article. We summarized the notations below to help reader understand.

## 2. Split-Hopkinson Pressure Bar (SHPB)

### 2.1. Theory

SHPB theory, which is based on the 1D wave equation, ignores friction and inertial effects between the pressure bar and specimen. Figure 1a is a schematic diagram of the SHPB, which is composed of a striker bar, incident bar, transmitted bar, and specimen. It is tested through the stress wave generated when the striker bar with a constant velocity ($V_{S}$) collided with the incident bar. The strain gauge is attached to the center of each pressure bar to avoid overlapping the elastic wave.

Figure 1b shows the X-T graph for the stress wave after the striker bar impact. The stress wave is generated immediately after the impact is propagated through the incident bar. The pulse measured by a strain gauge attached to the incident bar is called the incident pulse $\left(\varepsilon_{I}\right)$. When this incident pulse reached the specimen, an impedance difference between the pressure bar and specimen occurs. Impedance (Z=ρAC) is a function of the density of the material (ρ), elastic wave velocity (C), and area (A). When a wave is propagated from a medium with a larger impedance to a medium with a smaller impedance, a part of the wave is reflected, and the rest is transmitted. The reflected pulse $\left(\varepsilon_{R}\right)$ is measured on the strain gauge of the incident bar, and the transmitted pulse $\left(\varepsilon_{T}\right)$ is measured on the strain gauge of the transmitted bar. The specimen is compressed by the incident pulse, and governing equations such as the one-dimensional (1D) wave equation and force equilibrium is applied.
(1)$$\sigma_{S}\left(t\right)=\frac{A_{B}}{A_{S}}E_{B}\varepsilon_{T}\left(t\right)$$
(2)$$\varepsilon_{S}\left(t\right)=-2\frac{C_{B}}{L_{S}}\int_{0}^{t}\varepsilon_{R}\left(t\right)dt$$
(3)$${\dot{\varepsilon }}_{S}\left(t\right)=-2\frac{C_{B}}{L_{S}}\varepsilon_{R}\left(t\right)$$

In the test, the stress of specimen $\left(\sigma_{S}\right)$, strain of specimen $\left(\varepsilon_{S}\right)$, and strain rate of specimen $({\dot{\varepsilon }}_{S}$) could be obtained using the incident pulse $\left(\varepsilon_{I}\right)$, reflected pulse $\left(\varepsilon_{R}\right)$, and transmitted pulse$\left(\varepsilon_{T}\right)$ measured at each pressure bar, as expressed by Equations (1)–(3). $\sigma_{S}$, $\varepsilon_{S}$, ${\dot{\varepsilon }}_{S}$, $\varepsilon_{I}$, $\varepsilon_{R}$ and $\varepsilon_{T}$ are function of time (t). $A_{B}$ is area of pressure bar and $A_{S}$ is area of specimen. $C_{B}$ is the elastic wave velocity of pressure bar and $L_{S}$ is the length of specimen (=thickness) [18]. Figure 2 shows SHPB owned by the Department of Aerospace Engineering at Pusan National University.

To satisfy the 1D wave equation, the SHPB put the pressure bars and the specimen in a straight line, and the ratio of the length $\left(L_{B}\right)$ and diameter $\left(D_{B}\right)$ of the bar is more than 20 [19]. Table 1 shows the pressure bar length of the equipment used in this study.

The SHPB test uses reflection and transmission that result from the impedance difference of the stress wave, such that it is essential to know the impedance of the specimen and pressure bars. As the pressure bars, used in the SHPB, must behave elastically during the test, high-strength steel is used. In the case of stress waves, reflections and transmissions occur when propagating from a medium with high impedance to a medium with low impedance. When the impedance difference is reduced, the transmission rate increases. When the impedance passes through the same medium, complete transmission occurs. Therefore, if the impedance of the striker bar and incident bar is same, the stress wave generated in the striker bar is completely transmitted to the incident bar. Conversely, as the impedance difference increases, the reflection ratio increases, and the transmission ratio decreases [15]. When a pressure bar made of high-strength steel material is used to test a specimen such as urethane, which has a relatively low impedance, a low transmitted pulse may be obtained. It may cause difficulty in obtaining the material properties. To lower the impedance of the pressure bar, aluminum, titanium, and magnesium were used as materials in previous studies. Therefore, when the material of specimen is selected, it is important to choose the material of the pressure bar correspondingly. In this study, an Al6061-T6 specimen was used, and high-strength steel (SNCM439) was used as the pressure bar (Table 2).

### 2.2. Experiment

For each specimen, the diameter was the same (10 mm) and specimens of 1.5 t, 2 t, 3 t, 5 t, and 7 t, were tested to determine the influence of thickness on the acquisition of material properties at the same applied pressure (1.0 bar) (Figure 3). We performed the test for each thickness under the 1.0 bar test condition to understand the phenomenon that occurs when the thickness is changed under the same applied pressure condition, and Figure 4 shows the S-S curve accordingly. The S-S curve was increased as the thickness decreased during the test for the same applied pressure (Figure 4). According to previous studies, the lower the thickness, the higher the S-S curve because the effect of friction between the specimen and the pressure bar. Influence of friction is an important factor in selecting the specimen size. However, it is difficult to conclude that it is merely an increase due to the influence of friction because a high strain rate was also measured for the lowest-thickness specimen.

There is an experiment limit to understand the behavior of the test in real time because it was tested in milliseconds. Therefore, to discuss the deformation of the specimen, the finite element analysis program, LS-DYNA, was used to grasp the influence of thickness on the experiment.

### 2.3. Finite Element Method (FEM)

The analytical models are consisted of four parts: The striker bar, incident bar, transmitted bar, and specimen. Each part is the shape of a simple cylinder. The numerical values of each part reflect the actual dimensions (Table 3).

In the analysis model, it is important not only to perform accurate analysis, but also to reduce the analysis time. In the actual apparatus, the striker bar is designed to be accelerated in a pneumatic compression launcher to obtain the required velocity just before the impact. However, the distance between the striker bar and incident bar was 0.5 mm in analysis model because a constant velocity was imposed on the striker bar. The central axis of each part was placed on the same axis because the pressure bars and the specimen should be aligned in a straight line. Figure 5 shows the SHPB analysis model constructed in consideration of the above conditions. LS-DYNA can give the corresponding properties for each part by inputting the material card embedded in the program. Since the pressure bars behave in the elastic region in the experiment, there is no need for material properties for the plastic region. The material properties of the pressure bars were implemented using ‘MAT_001_ELASTIC’ material card. However, the specimen was made of Al6061-T6, and ‘MAT_024_PIECEWISE_LINEAR_PLASTICITY’ material card was used to implement the plastic region. The ‘MAT_024_PIECEWISE_LINEAR_PLASTICITY’ material card basically requires data of density, Young’s modulus, Poisson’s ratio and yield strength (Table 4).

The analysis on the plastic region can be implemented by inputting curves considering strain rate. Therefore, the experiment was performed under the test conditions of 0.5 bar, 1.0 bar, and 1.5 bar with Al6061-T6 specimen of 5 t, and a curve fitting was performed by applying the simplified Johnson-Cook(J-C) model [18] with no temperature consideration:(4)$$\sigma =\left(A+B\varepsilon^{n}\right)\left(1+Cln\dot{\varepsilon }\right).$$

In this equation, $\sigma \;(N/m^{2}$) is the equivalent stress, ε (mm/mm) is the equivalent plastic strain, and $\dot{\varepsilon }\;\left(s^{-1}\right)\;$is the dimensionless strain rate. The material constants are A, *B*, n, C. A is the yield stress of the material, *B* is the strain hardening constant, n is the strain hardening coefficient, and C is the strengthening coefficient of strain rate [26]. Table 5 is a parameter obtained by applying Equation (4) based on quasi-static data and SHPB experimental data. Figure 6 shows quasi-static data, J-C model data for each strain rate, and S-S curves of 3 t, 5 t, and 7 t specimens obtained under 1.0 bar test conditions.

In the SHPB test, a pneumatic compression launch device was used to impart a constant velocity to the striker bar. Tests for each thickness were performed at 1.0 bar. Due to the acceleration of the striker bar fired from the pneumatic compression launcher, it was impossible to obtain a constant velocity. Therefore, the velocity just before the striker was taken as the initial velocity for the analysis, and 17.5 m/s was used according to the experimental value. In addition, a non-frictional analysis was performed because no friction between the specimen and the pressure bars was assumed.

## 3. Influence of Thickness

### 3.1. Transmitted Pulse

The SHPB test was performed using stress waves generated by the striker bar colliding with the incident bar during the test. In other words, obtaining different material properties when testing for each thickness under the same applied pressure means that the stress wave is different. The stress wave is acquired from the strain gauge of each pressure bar, denoted as the incident pulse $\left(\varepsilon_{I}\right)$, reflected pulse $\left(\varepsilon_{R}\right)$, and transmitted pulse $\left(\varepsilon_{T}\right)$. First, the incident pulse $\left(\varepsilon_{I}\right)$ is only affected by the initial impact velocity between the striker bar and incident bar. The thickness of the specimen does not affect the waveform. Therefore, only the transmitted $\left(\varepsilon_{T}\right)$ and reflected pulses $\left(\varepsilon_{R}\right)$ were compared for each thickness to determine the influence of thickness. Figure 7 is a comparison of the measured transmitted pulses when the analysis for each thickness is performed under the same velocity condition of 17.5 m/s. For the transmitted pulse, as the thickness of the specimen decreased, the peak value of the wave pulse increased, as shown in Figure 7. The transmitted pulse measured during the test derives the stress value of the specimen using Equation (1). The difference in the transmitted pulse according to the thickness at the same applied pressure implies that different material properties are acquired. As such, the lower the thickness, the higher the stress value according to the increase of the transmitted pulse during the test.

The impedance of the specimen was analyzed to understand the mechanism of the change in the transmitted pulse with thickness under the same conditions. Since the SHPB operates via the propagation of the stress wave, the impedance of the medium through which the stress wave is propagated is an important factor. The elastic wave velocity is expressed as $C=\sqrt{E/\rho }$. Since the density and elastic wave velocity are inherent material properties, they are constant. Therefore, the impedance can be expressed as Z=f(A) and is function of area. The impedance of the transmitted pulse increases as the thickness decreases, for the same incident pulse. The analysis assumed that there was no friction between the bar and the specimen, and the specimen was compressed evenly when deformed. These assumptions differ from the actual test conditions, but the magnitude of the area change in the analysis should not be ignored.

Figure 8 shows the area and impedance values of the specimen compressed by elastic waves under the same applied velocity condition of 17.5 m/s for each thickness. The smaller the thickness, the larger the area change. The strain value in the thickness direction was measured to be higher after data processing (Figure 8a). At the same applied pressure, the lower the thickness, the higher the impedance value of the specimen. The amount transmitted from the same incident pulse to the transmitted bar increases (Figure 8b). The transmitted pulse differs for each thickness under the same applied pressure because the impedance varies according to the change in the area of the specimen.

### 3.2. Reflected Pulse

Figure 9 is a comparison of the measured reflected pulses when the analysis for each thickness was performed under the same velocity condition of 17.5 m/s. As the thickness of the reflected pulse decreases, the peak of the stress wave decreases (Figure 9). In SHPB theory, the incident pulse is divided into transmitted and reflected pulses and can be expressed by Equation (5) [18].
(5)$$\varepsilon_{I}=\varepsilon_{T}+\varepsilon_{R}$$

In this equation, $\varepsilon_{I}$ is the incident pulse, $\varepsilon_{R}$ is reflected pulse, and $\varepsilon_{T}$ is transmitted pulse measured at each pressure bar. The reflected pulse decreases as the shape of the transmitted pulse increases as the thickness decreases. The stress and strain rate of the specimen were obtained from Equations (2) and (3) through the reflected pulse. In Figure 9, as the thickness decreases, the peak of the reflected pulse decreases. We predicted that the resulting values of strain and strain rate decreases. However, as shown in Figure 8, the larger area change rate for the thin specimen implies that the strain and strain rate were high and this causes contradictions. Therefore, the strain-strain rate curve was analyzed to understand the relationship between the reflected pulse, strain, and strain rate for each specimen.

### 3.3. Strain-Strain Rate Curve

Figure 10 shows the strain and strain rate obtained through Equations (2) and (3) for each thickness using the reflected pulse of Figure 9. As the thickness decreases, the resulting values of strain and strain rate increases, as shown in Figure 10. This is the opposite result to the previous decrease in the reflected pulse. In Equations (2) and (3), as the thickness decreases, the reflected pulse decreases. However, the ratio of the initial specimen length ($L_{0}$) to the reflected pulse reduction ratio is dominant and increased the strain and strain rate. This is the same result as predicted in Section 3.1, where the strain and strain rate increased as the area increased under the same test conditions. That is, when specimens of different thicknesses are used at the same applied pressure, the deformation is occurred in different strain rate regions.

During the SHPB test, the specimen must be deformed within a uniform strain rate. As the thickness decreases, the strain rate region fluctuated more severely, resulting in poor reliability even when tested. As the thickness decreases, testing in the high-strain-rate region at the same applied pressure becomes possible while the reliability of the data decreases.

### 3.4. Stress-Strain Curve

Figure 11 shows the stress and strain rate obtained through Equation (1) for each thickness using the transmitted pulse of Figure 7. The data of 1.5 t and 2 t were excluded. They were judged to have low reliability under the SHPB test conditions because the strain rate fluctuated severely. As the thickness decreases, the stress and strain values are increased. In Section 3.3, the strain rate varied depending on the thickness. The lower the thickness, the more deformed the specimen in the high-strain-rate region. However, from the above results alone, it was difficult to determine whether the increased curve was due to thickness or other effects. Therefore, further analysis was performed under the condition of initial velocity ($V_{S}$) that created a specific strain rate region for each thickness.

### 3.5. Different Initial Velocity Condition

In Section 3.4, deformation occurred in different strain rate regions under the same applied pressure conditions for each thickness. Therefore, different applied pressures were imposed in the 3 t, 5 t, and 7 t analytical models. The results were compared when the strain was performed in the similar strain rate region. To compare the results of the 1.0 bar test of the 5 t specimen, the following pressures were used: 0.5 bar (12.5 m/s) for 3 t, 1.0 bar (17.5 m/s) for 5 t, and 1.5 bar (22.5 m/s) for 7 t. Figure 12 shows the strain and strain rate curves obtained by using the reflected pulses acquired in the above test conditions by thickness. For different applied pressures, deformation occurred within a range of similar strain rates (Figure 12). The variation in strain rate increases as the thickness decreases. It causes a small error during the acquisition of the material properties. The thinner the thickness, the more sensitive the strain rate change. This should be considered in choosing the thickness of the specimen. Figure 13a shows the analysis results under the above test conditions by thickness. Figure 13b shows the experiment results. In Figure 13, when the applied pressure that creates a similar strain rate region for each thickness was used, there is no difference observed in the S-S curve. That is, the decrease in the thickness under the same applied pressure showed an increase in the linearity, which was not due to the influence of friction, but due to the deformation in the high-strain rate region. The S-S curve obtained from the SHPB is an inherent property of the material.

## 4. Conclusions

To perform the SHPB test, selecting an appropriate specimen size is important. Although the specimens of the quasi-static test have ASTM standards, the specimens of the SHPB test have been used without specific standards. In previous studies [27], the cause of the increase in the S-S curve when the thickness decrease is often identified as the influence of friction. However, it is important to understand the influence of specimen thickness on experiment. This study investigated the effect of the thickness of the specimen on the material properties by varying the thickness of an Al6061-T6 specimen under the same applied pressure. Since there is an experimental limit to understand the phenomenon of thickness, it was verified using the FEM program, LS-DYNA.

Conclusion 1: The reason of the S-S curve variation by thickness of the specimen under the same pressure is the effect of different strain rates depending on the thickness of the specimen. Under the same pressure conditions, as the thickness of the specimen increased, the peak of the transmitted pulse increased. Through the analysis, it was confirmed that the area change of the thin specimen was larger than that of the thick specimen. Also, the transmitted pulse increased through the transmission of more stress waves due to the increase in impedance. In the case of the reflected pulse, despite the decrease of peak point, a high strain rate region was created under the influence of the thickness value itself. In conclusion, the lower the thickness, the higher the S-S curve.

Conclusion 2: Different thickness of the specimen does not affect the acquisition of material properties if the test is performed in the similar strain rate. However, the thinner the specimen, the greater the variation in the strain rate curve. This may lead to reliability problems in obtaining the material properties. Therefore, it is not favorable to significantly reduce the thickness of the specimen. In this study, the analysis was performed without the effect of friction. In actual SHPB test, friction effect cannot be ignored. It is expected that more reliable physical properties can be obtained if further research on friction due to thickness reduction are made.

## Figures and Tables

**Figure 1 materials-14-00205-f001:**
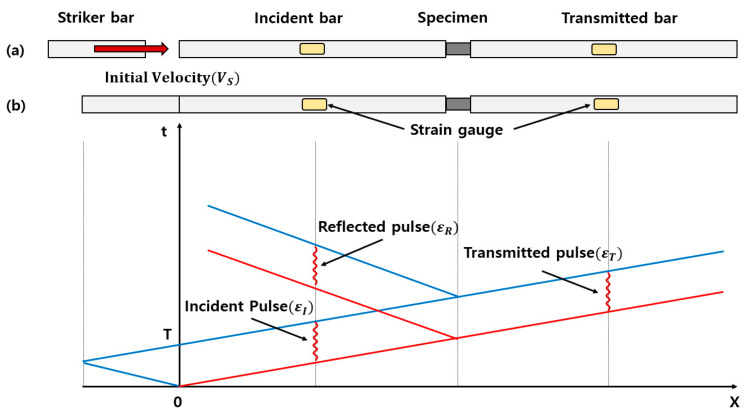
(**a**) Schematic diagram of an SHPB; and (**b**) the SHPB wave propagation after the impact of the striker bar [18].

**Figure 2 materials-14-00205-f002:**
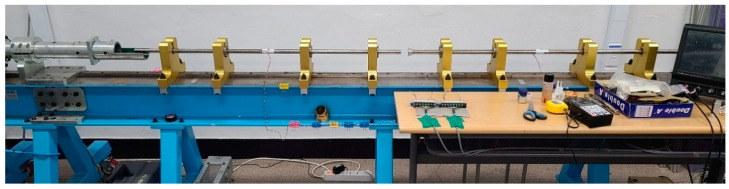
SHPB used in this study, owned by the Department of Aerospace Engineering at Pusan National University [25].

**Figure 3 materials-14-00205-f003:**
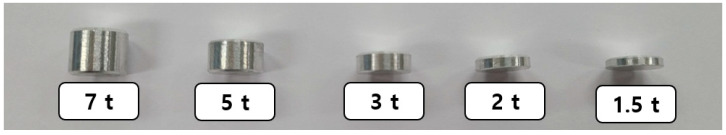
Al6061-T6 cylinder specimen of different thicknesses (1.5 t, 2 t, 3 t, 5 t, and 7 t).

**Figure 4 materials-14-00205-f004:**
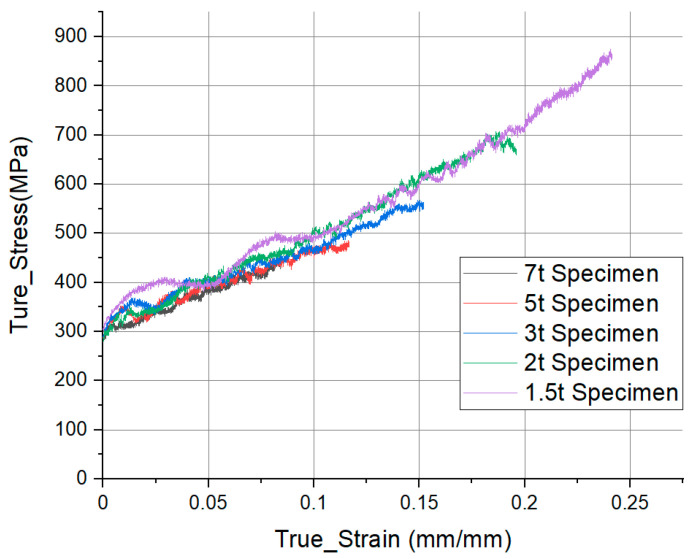
Experiment data: True strain and stress (S-S) curve for specimens of different thicknesses in 1.0 bar condition.

**Figure 5 materials-14-00205-f005:**
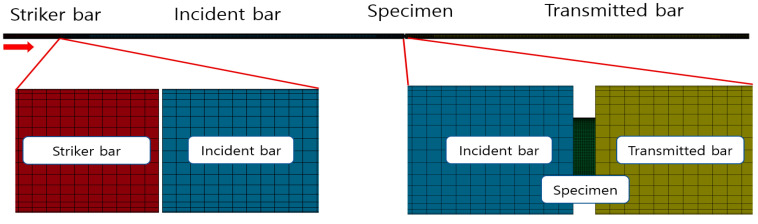
SHPB apparatus FEM modeling in LS-DYNA.

**Figure 6 materials-14-00205-f006:**
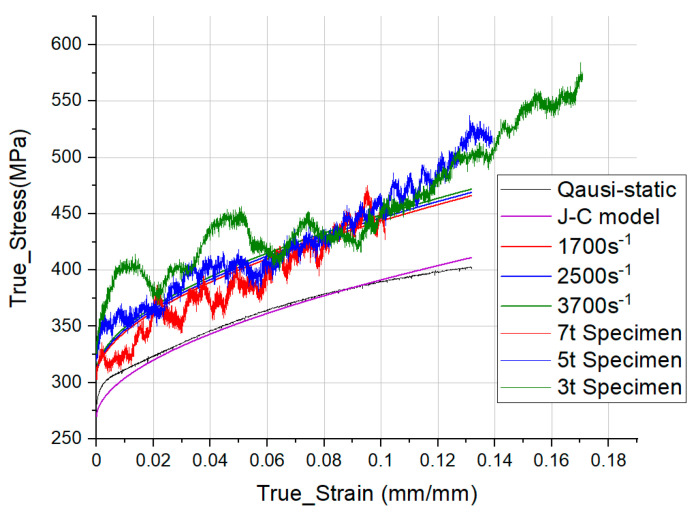
Johnson-Cook fitting model of the Al6061-T6 specimen with the SHPB experimental results.

**Figure 7 materials-14-00205-f007:**
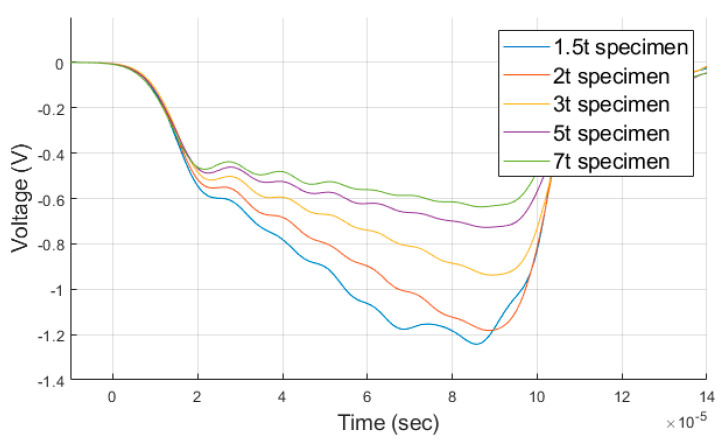
Comparison of the transmitted pulse waveforms, measured in the analysis for each thickness.

**Figure 8 materials-14-00205-f008:**
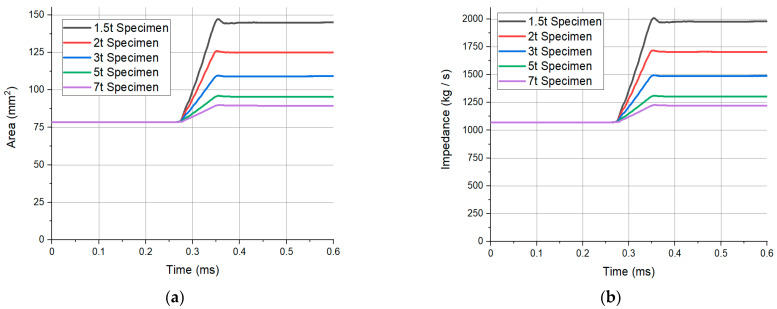
Real-time, specimen; (**a**) area; and (**b**) impedance graph obtained from the analysis of each thickness.

**Figure 9 materials-14-00205-f009:**
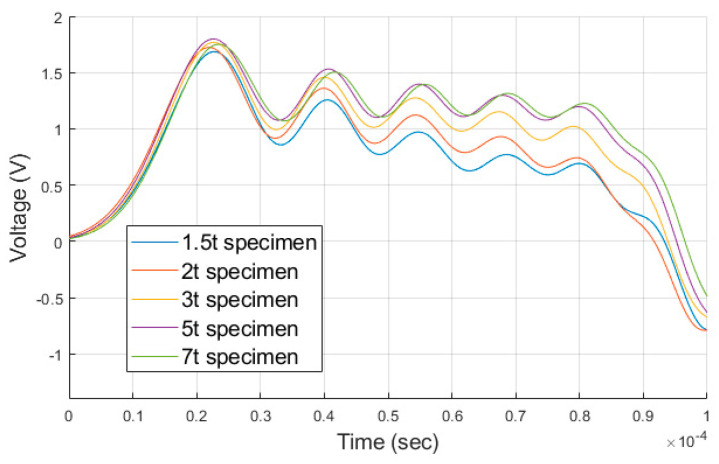
Comparison of the reflected pulse waveforms, measured in the analysis for each thickness.

**Figure 10 materials-14-00205-f010:**
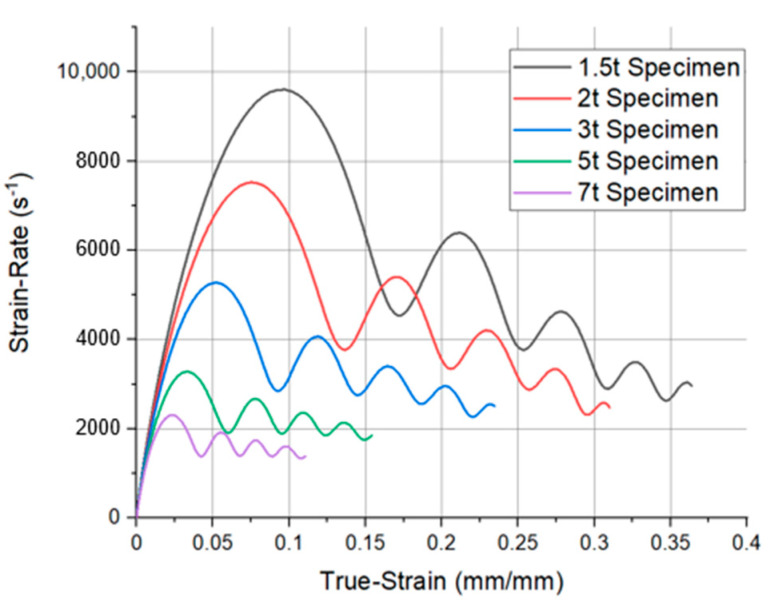
Comparison of true strain–strain rate graph, measured in the analysis for each thickness.

**Figure 11 materials-14-00205-f011:**
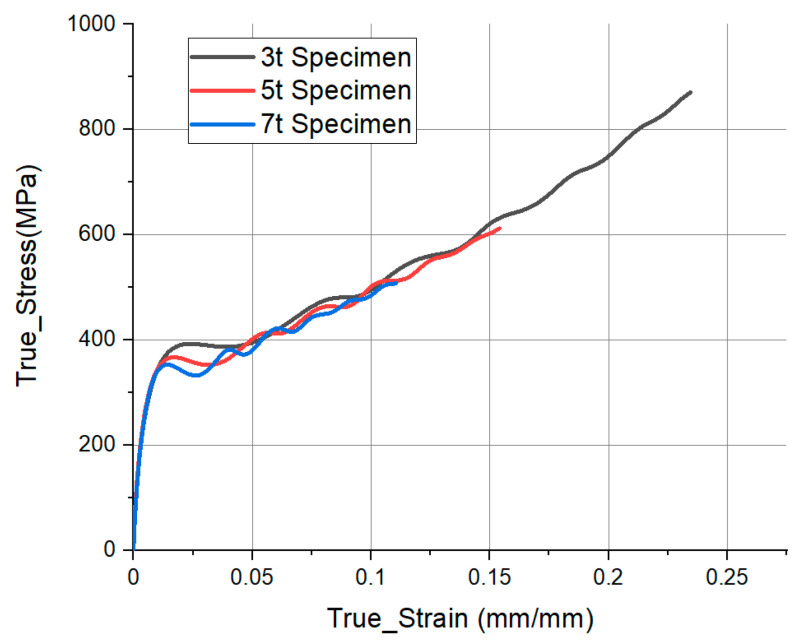
Comparison of true stress–strain graph, measured in the analysis for each thickness.

**Figure 12 materials-14-00205-f012:**
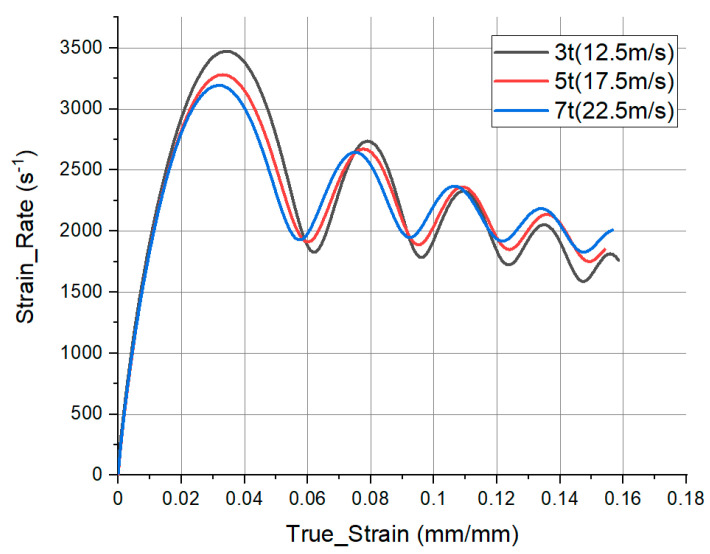
Comparison of the strain and strain rate results under different test conditions.

**Figure 13 materials-14-00205-f013:**
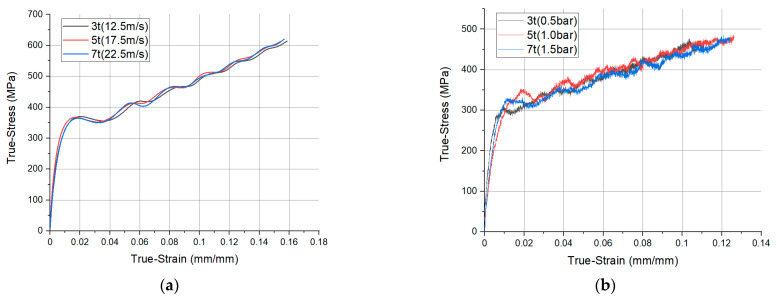
Comparison of the true stress and strain results in; (**a**) analysis; and (**b**) experiment under different test conditions.

**Table 1 materials-14-00205-t001:** Dimensions of the SHPB pressure bar.

	Length (mm)	Diameter (mm)
Striker bar	200	20
Incident bar	1200	20
Transmitted bar	1200	20

**Table 2 materials-14-00205-t002:** Material properties of SNCM439.

Material: SNCM439	
Density (ρ)	7850 kgm^−3^
Young’s modulus	196 (GPa)
Poisson ratio	0.3
Yield strength	1550 (MPa)

**Table 3 materials-14-00205-t003:** The dimensions of each part in FEM modeling.

	Striker Bar	Incident Bar & Transmitted Bar	Specimen
Length	200 mm	1200 mm	1.5/2.0/3.0/5.0/7.0 mm
Diameter	20 mm	20 mm	10 mm

**Table 4 materials-14-00205-t004:** Material properties of Al6061-T6.

Material: Al6061-T6	
Density (ρ)	2700 kgm^−3^
Young’s modulus	68.9 (GPa)
Poisson ratio	0.33
Yield strength	270 (MPa)

**Table 5 materials-14-00205-t005:** Johnson-Cook model parameters of Al6061-T6.

A(MPa)	B(MPa)	n	C
270	430	0.55	0.018

## Data Availability

The data presented in this study are available on request to the corresponding author.

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
