# Peer review of "Influence of Specimen Thickness on the Acquisition of Al6061-T6 Material Properties Using SHPB and Verified by FEM"

_materials, 2021, doi:10.3390/ma14010205_

Round 1

Reviewer 1 Report

  • The current study aims to extract material properties of Al6061 aluminium alloy using split Hopkinson pressure bar. The objective here is to understand how the thickness of the specimen influence the obtained material properties. The authors also develop a numerical model using a commercial FE software to simulate the stress strain curves obtained from the experimental results. The study also investigates the influence of pressure level on the strain rates of the tested material. The authors reported findings from the tests and concluded that thickness change does not impact the overall mechanical properties of the tested material.
  • Line 12-13 why the text is greyed? Same in lines 16-17, please check
  • Line 31 “high speed situation…” please revise this sentence. Perhaps use word such as event or application ….etc
  • Line 31 just say a low or high-speed impact incidents such as an aircraft or a car crash…
  • Line 40 please revise the sentence wording
  • The authors are strongly recommended to carry out full English editing and proofing as a lot of errors and incorrect wording appears to be in the manuscript.
  • The literature review is short, the authors are encouraged if they can find past work on lab made developed SHPB for testing different materials and what does their home made devices differs from the traditional ones and from the ones you are using in this work.
  • Please reference Table 2
  • Section 2 in the paper can be reduced, it is too descriptive of how the SHPB works and perhaps very detailed description of how it operates is unnecessary. The authors are encouraged to just briefly explain how the test was set and what does their device provides differently from other SHPB.
  • Table 3 is missing reference from did the authors get the JC parameters. Or is this data obtained from your own experimental tests? Also for JC properties, many authors have obtained different set of values using their own SHPB tests. Have the authors considered testing more than one set of data from different sources? Check the following for example:
  • https://www.sciencedirect.com/science/article/pii/S1877705811007612
  • https://journals.sagepub.com/doi/full/10.1177/1687814018797794
  • https://link.springer.com/article/10.1007/s40870-017-0096-4
  • https://core.ac.uk/download/pdf/76989889.pdf (DEVELOPMENT OF HIGH STRAIN RATE MECHANICAL TESTING FOR METALLIC MATERIALS)
  • the authors are encouraged to combine figure 7-10 to make it easier for the readers to follow your discussion throughout the manuscript.
  • there is no discussion at al about the FE model used in the study, the authors need to discuss this and explain how they developed this model and what are boundary conditions, setup and loads applied…etc.
  • in line 196-207: is there is a minimum thickness for samples to be used in SHPB after which any lower would not result in correct representation of the behaviour of the material under certain strain rates?
  • Also how many times were each tests with specific parameters was repeated? If each test was carried out one time this might not give accurate results.
  • Line 211, perhaps there is a minimum thickness to be used for certain strain rates (impact speeds) applied in the tests.
  • Line 228 yes but is this difference in decrease due to change in thickness is significant, as you mentioned after this line you indicate the possibility of errors.
  • The authors are encouraged to combine figures 13-15.
  • For Literature review the authors are encouraged to create a table that summarised past attempts on studying Al6061 or other Al alloys using SHPB and what were the test conditions and what was evaluated from each study to better reflect on previous work done in this field.

Author Response

We sincerely appreciate your reviews and comments. We have looked closely at your comments. Please see the attachment.

Reviewer 2 Report

The reviewer comments of the paper «Influence of thickness on the material properties of an Al6061-T6 specimen using a Split-Hopkinson pressure bar»

- Reviewer

The authors presented an article «Influence of thickness on the material properties of an Al6061-T6 specimen using a Split-Hopkinson pressure bar». However, there are several points in the article that require further explanation.

Comment 1:

On the one hand, the abstract is written correctly. However, demonstrate in the abstract novelty, practical significance. Add quantitative and qualitative work results to the abstract.

Comment 2:

Overall, the introduction is well written. However, explain why the material chosen for research is so important for the study. Provide a paragraph with relevant references for this material. Clearly identify "white spots". That is, what will be done in this work. Be clear about the novelty of the research. Analyze in the introduction the articles on applying the split Hopkinson test to the Al6061-T6, for example: doi: 10.1016/j.matchar.2017.03.005 ; doi: 10.1016/j.msea.2019.02.016

Also in the introduction it will be useful to add a paragraph analyzing previous work on the topic of the article using the finite element method. And here it is important to show what is the difference in the approach proposed in the article. It is useful to add an article: doi: 10.1007/s11029-015-9478-7

At the end of the introduction, provide a clear and understandable purpose of the research you are doing.

Comment 3:

Section 2 should be used. «2. Materials and methods» Sections:

2.1 Theory

2.2 Experiment

2.3 Finite element method

Use full section names instead of abbreviations: 2. SHPBs 2.1. SHPB theory

Describe the measurement procedure in detail. The article contains 15 figures with different physical quantities. All this should be clearly and clearly described in Section 2 Materials and Methods.

Comment 4:

Explain the mathematical models described in article are original and suggested by the authors? In addition, all physical quantities included in the equations must be listed and deciphered after each formula. All images borrowed must be obtained by the appropriate permision from the publisher.

The article is devoted to the studies of Al6061-T6, and Table 2 considers Material: SNCM439. Everything should be in accordance with the ongoing research. What is the hardness of Al6061-T6?

Provide a design scheme for FEM analysis.

What materials is each unit made of? Was this taken into account when creating the FEM model?

What are the boundary conditions?

The FEM model itself, the selected type of finite elements, the inhomogeneity of the finite element mesh should be described in detail and justified in the text of the article.

Give PC characteristics, the name of the soft used for FEM. Justify the choice.

Comment 5:

Show photos or SEM of samples of different thickness before and after rupture. It is important to show on which surfaces the sample is destroyed. Does the fracture surface microstructure differ for dimensions? It is useful to discuss this in the article.

Comment 6:

What are the parameters in Table 3? This should all be clearly and clearly described and explained. There is a reference to table 2, but it does not reflect the properties of Al6061-T6.

Figures 5, 6, 7, 8, 9, 10, 11, 12, 13, 14, 15 in the article should be analyzed and described in more detail and clearly. Now the text for these figures is not able to convey anything concrete to the reader.

Comment 7:

It will be useful to add a section of Nomenclature in which to sign all the physical quantities and abbreviations encountered in the article. There are many physical quantities in the text and such a section will help to find the description of the necessary element.

For example,

?0           : Initial specimen length

FEM      : Finite Element Method

etc.

Comment 8:

Conclusions

Use the format:

  • Conclusion 1
  • Conclusion 2

etc.

Add quantitative and qualitative work results.

In addition, it is necessary to more clearly show the novelty of the article and the advantages of the proposed method. What is the difference from previous work in this area? Show practical relevance. Conclusions should reflect the purpose of the article.

Comment 9:

Title's articles should also be changed. It should more clearly reflect the purpose of the article and include the key methods used in the article and materials, in particular FEM.

Comment 10:

The English in the article should be significantly improved.

The topic of the article is interesting, but in its current form, the article cannot be suitable for publication in a prestigious international journal. Now it is more like a short report without a clear and understandable analysis of the results. However, authors should carefully study all comments. Only after major changes can an article be considered for publication in the "Materials".

Author Response

(The authors gave the same response as above.)

Round 2

Reviewer 1 Report

All questions have been answered, the reviewer suggest to improve the literature more and ciritically discuss past work on similar topic to increase the visibility of the paper in the scientific and industrial communities.

Author Response

We sincerely appreciate your reviews and comments. Please see the attachment.

Reviewer 2 Report

The authors considered the comments superficially. Therefore, the revision of the article did not take into account all of the indicated comments. In this regard, it is necessary to prepare a new revision of the article where in detail, step by step, there will be an answer to all comments, and not selectively.  

And it is also important that after each formula the physical quantities encountered for the first time should be explained. All physical quantities and abbreviations must be added to the nomenclature.  

The reference list must be fulfilled according to the MDPI requirements.  

Therefore, a major revision is needed.

Author Response

(The authors gave the same response as above.)

Round 3

Reviewer 2 Report

Comment 2: And it is also important that after each formula the physical quantities encountered for the first time should be explained. This comment must be completed as required. These are the ethics and requirements of scientific publication. However, in the nomenclature section, let the same values be listed as before.

Author Response

We sincerely appreciate your quick reply. Please see the attachment.
